# SpecMD: A Comprehensive Study On Speculative Expert Prefetching

**Duc Hoang** [1]   **Mohammad Samragh** [1]   **Ajay Jaiswal** [1]   **Minsik Cho** [1]

## Abstract

Mixture-of-Experts (MoE) models enable sparse expert activation, meaning that only a subset of the model's parameters is used during each inference. However, to translate this sparsity into practical performance, an expert caching mechanism is required. Previous works have proposed various caching policies, but each evaluates only 1–2 policy combinations in isolation, leaving cross-policy interactions poorly understood. To address this gap, we develop **SpecMD**, a standardized benchmark for evaluating training-free, drop-in MoE cache policies under controlled hardware constraints. Using SpecMD, we perform an exhaustive benchmarking of several MoE caching strategies, reproducing and extending prior approaches in controlled settings with realistic constraints. Our experiments reveal that MoE expert access is not consistent with temporal locality assumptions (e.g LRU, LFU). Motivated by this observation, we propose **Least-Stale**, a novel eviction policy that exploits MoE's predictable expert access patterns to reduce collision misses by up to $85\times$ over LRU. With such gains, we achieve over $88\%$ hit rates with up to $34.7\%$ Time-to-first-token (TTFT) reduction on OLMoE at only $5\%$ or $0.6GB$ of VRAM cache capacity.

## 1. Introduction

Mixture-of-Experts (MoE) architectures scale language models to unprecedented sizes through sparse computation. By activating only $k$ of $N$ experts per token, MoE models like Mixtral-8x7B (Jiang et al., 2024), DeepSeek-V2 (DeepSeek-AI et al., 2024), and OLMoE (Muennighoff et al., 2025) achieve sub-linear computational cost while maintaining large model capacity. However, this efficiency comes at a cost: a naive deployment of MoE models requires

[1]Apple. Correspondence to: Duc Hoang <dhoang2@apple.com>.

*Proceedings of the $43^{rd}$ International Conference on Machine Learning*, Seoul, South Korea. PMLR 306, 2026. Copyright 2026 by the author(s).

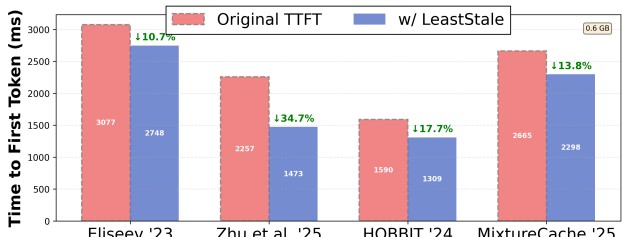

*Figure 1.* Least-Stale eviction as drop-in improvement across baseline approaches. Our novel eviction policy achieves 10.7–34.7% TTFT reduction on OLMoE-1B-7B at 5% cache capacity (0.6 GB).

all $N$ experts to remain memory-resident, making deployment prohibitively expensive. For instance, assuming 16 bits per weight, OLMoE-1B-7B activates 2.39GB yet needs 12.0GB ready in memory. Mixtral-8x7B requires 88GB storage yet only uses 24GB. This linear memory scaling limits the MoE deployment on memory-constrained devices.

Instead of keeping the full MoE model in active memory, an alternative is to store only the subset of patronized experts in a fast but capacity-limited cache, making memory management central to performance which can be influenced by the four aspects: •**routing**: how to select experts to be cached, •**prefetching** how to predict and cache the experts needed in future, •**eviction** how to free space in the cache for demanded experts, and •**miss handling** how to respond when required experts are absent from the cache. The importance of the above four aspects varies depending on hardware specifications. For instance, routing is capacity-dependent, prefetching is constrained by bandwidth, eviction matters in low-capacity settings, and miss handling is critical in balancing user-experience and task-performance.

Existing approaches tackle the cache management challenges through routing modification (Skliar et al., 2025), LRU caching (Eliseev & Mazur, 2023), and dynamic precision (Tang et al., 2024; Zhu et al., 2025), but each evaluates only 1–2 policy combinations under specific hardware assumptions, leaving general insights into policy interactions underexplored. This raises a fundamental research question: *when, where, and which combinations of these policies would offer the most benefits across different deployment scenarios?*

It is for this end that we develop **SpecMD**, a standardized benchmark designed to enable reproducible comparison of training-free MoE caching strategies. Through this benchmark, we gain three capabilities: (1) controlled comparison of prior approaches under identical conditions, (2) exploration of policy interactions and compositions, and (3) characterization of performance across the full hardware constraint space.

Using SpecMD, we conducted an exhaustive benchmarking study of MoE caching policies. We tested LRU caching (Eliseev & Mazur, 2023), fetch cascade (Tang et al., 2024), and expert substitution (Zhu et al., 2025) under controlled and comparable conditions. We evaluate four MoE models (Mixtral-8x7B, OLMoE-1B-7B, Phi-3.5-MoE, Qwen1.5-MoE) and three different hardware configurations. From our studies, we made the following observations:

- **MoE's expert access can cause wrongful eviction.** In MoE, experts are accessed following a deterministic sequential layer pattern, not recency-based reuse, rendering LRU and LFU policies perform quite poorly.

- **Dynamic prefetching is better for cache.** We observe dynamic, score-based prefetching outperforms top-k approaches; despite lower overall prediction accuracy, score-based prefetching yields higher hit-rates and makes use of available bandwidth more effectively.

- **Policy rankings shift across hardware regimes.** As expected, the best-performing strategy at 1% capacity differs from that at 5% or 25%; bandwidth-limited and capacity-limited scenarios favor different approaches. No single set of policy dominates, but there are trends (see Figure 9 in Appendix).

From these findings, we propose **Least-Stale**, an eviction policy that balances temporal and spatial awareness to minimize cache collision misses. Our contributions are:

- **A Benchmark for MoE Caching Policies.** We introduce an open-source benchmark that decomposes the MoE cache policy space into four orthogonal dimensions and enables reproducible, controlled comparison across models and hardware constraints.

- **Systematic Policy Study.** Using SpecMD, we conduct the first cross-policy interaction study across four MoE architectures and three constraint regimes, revealing key insights for practitioners.

- **Least-Stale Eviction Policy.** Building on our findings, we propose Least-Stale, a flexible eviction policy that greatly reduces collision-misses by up to $85\times$ versus LRU at 1% capacity, achieving 88-92% hit rates. We show in Figure 1 Least-Stale provides consistent gains across diverse baseline strategies.

## 2. Related Work

MoE memory constraints are addressed through several complementary approaches: routing modification, expert caching with prefetching and quantization. We focus on expert caching approaches, as they directly target cache management, i.e., how to decide which experts to load or evict and how to handle cache misses. Existing work evaluates 1-2 policy combinations in isolation, leaving policy interactions and performance across hardware constraints largely uncharacterized. A parallel line of work targets system-level throughput through pipelining and batching: MoE-Lightning (Cao et al., 2024) introduces a CPU-GPU-I/O pipelining schedule with paged weights, and MoE-Gen (Xu et al., 2025) accumulates tokens to launch large module-level batches that overlap computation with communication. These efforts are orthogonal to the policy-level decisions we study and can compose with any underlying caching policy.

**Prefetching.** Eliseev & Mazur (2023) introduce speculative prefetching for memory-constrained MoE inference. Their approach predicts next-layer expert requirements using current-layer hidden states, enabling single-layer-ahead speculation. HOBBIT (Tang et al., 2024) extends this by leveraging gating input similarity across adjacent layers to prefetch multiple layers ahead. Pre-gated MoE (Hwang et al., 2023) pursues a co-design route, training a pre-gating module that predicts the next layer's expert assignment to enable reliable lookahead at the cost of model modification. ExpertFlow (He et al., 2024) extends prediction further by training a transformer-based predictor that forecasts expert usage across all MoE layers in a single forward pass and pairs it with predictive caching and token scheduling. Klotski (Fang et al., 2025) folds a correlation-aware prefetcher into an expert-aware multi-batch pipeline, hiding I/O behind computation across batches. These approaches predict based on architectural or learned proximity but do not account for expert importance when bandwidth is limited. Zhu et al. (2025) addresses this through score-based prefetching, which filters experts by gate score percentile rather than fixed top-$k$ counts. This prioritizes high-importance experts when bandwidth cannot accommodate all predictions.

**Eviction.** Eviction policies determine which cached experts to remove when capacity is exceeded. Eliseev & Mazur (2023) employ LRU caching, keeping a fixed number of recently-used experts across all layers. Tang et al. (2024) proposes LHU (Least High-precision Used), a variant of LFU that prioritizes retaining high-precision experts. Zhu et al. (2025) use score-based eviction, removing experts with the lowest accumulated activation scores.

**Miss handling.** For cache misses, Eliseev & Mazur (2023) trigger blocking loads from CPU memory. HOBBIT introduces precision cascading: dynamically loading lower-precision versions (int4 for float16, int2 for int8) based on

gating score thresholds (Tang et al., 2024). Zhu et al. (2025) employ expert substitution, replacing cache-miss experts with functionally similar cached experts whose scores fall within tolerance thresholds.

**Routing.** MixtureCache (Skliar et al., 2025) takes a fundamentally different approach by modifying routing rather than optimizing cache policies. They bias expert selection toward already-cached experts through learnable parameters applied to router logits. Their system employs LRU eviction but no prefetching mechanism, instead treating cache misses as inevitable and modifying routing to reduce misses.

## 3. SpecMD: Benchmark Design and Policy Space

We present a benchmark that prioritizes ease of use (straightforward integration with HuggingFace models), accessibility (no specialized hardware required), and policy modularity without any training requirement. We aim to enable straightforward policy comparison across models and hardwares on consumer-grade GPUs[1].

### 3.1. Policy Design Space

Cache policies interact across four dimensions, shown in Figure 2, each affecting *quality*, *speed*, or both. Understanding these interactions is the central challenge. We systematically explore this design space to characterize policy behaviors and identify optimal configurations under different quality-memory constraints.

#### 3.1.1. ROUTING POLICIES

When an expert is requested by the gate module in an MoE block (top left of Figure 2), it may not exist in the cache. In this case, the routing policy is the first module that we may visit. Routing policies (shown as a green policy box in the Main Loop of Figure 2) determine whether to influence expert selection based on cache state, impacting both *performance* and *speed*. This represents a trade-off between cache efficiency and routing integrity.

**Standard routing** preserves original model behavior using unmodified router logits, maintaining 100% routing fidelity. All other policy dimensions operate around the model's original expert selections.

**Cache-aware routing** biases selection toward cached experts by boosting GPU-resident expert logits (Skliar et al., 2025). Let $z = G(x)$ represent original router logits, $C$ denote cached expert indices, and $\mathbb{1}_C \in \{0,1\}^{|E|}$ indicates cached experts. Modified logits are

$$z' = z + \lambda \cdot \Delta_{avg} \cdot \mathbb{1}_C, \tag{1}$$

where $\lambda$ is hyperparameter used for tuning and $\Delta_{avg}$ is the historical average logit value. Modified logits $z'$ pass through softmax and top-$k$ selection as usual.

#### 3.1.2. CACHE MISS POLICIES

After the routing policy is applied, the expert may still not be available in cache. Cache miss policies, as the name suggested, handle missing experts from cache (this is showed in the Main Loop of Figure 2). When cache misses occur, these policies can force immediate eviction to make space and may stall the model until the needed experts are fetched, substituted, or dropped. Unlike eviction/prefetching (primarily affecting speed), miss handling impacts both **quality** and **speed**. We implement three cache miss policies:

**Fetch policies** always fetch missing experts from CPU memory, preserving routing integrity at the cost of synchronous transfer latency. Basic Fetch guarantees 100% fidelity but degrades speed when misses are frequent. We introduce two precision-aware variants: *Fetch-lowest-bit* (e.g., int8:4) fetches the lowest available precision to minimize latency, while *Fetch-priority* (e.g., int8:4:2) always fetches highest precision with fallback levels (int8→int4→int2) for graceful degradation under capacity constraints.

**Drop policies** selectively skip cache-miss experts. Skliar et al. (Skliar et al., 2025) show MoE models are robust to expert variations: dropping top-ranked experts compromises performance, but models are resilient beyond rank 2, especially granular MoEs with many active experts. They demonstrate minimal perplexity degradation when swapping rank 3+ experts, which we leverage similarly by dropping only low-rank experts and always fetching high-rank ones.

**Substitution policies** replace cache-miss experts with functionally similar cached experts based on score proximity (Zhu et al., 2025), but quality degrades significantly in practice (Section 5.3).

#### 3.1.3. EVICTION POLICIES

After the cache-miss policy is applied, an expert may need to be loaded into the cache. Eviction policies determine which experts to remove when cache capacity is exceeded, and play a fundamental role in both the Main and Watchdog loop as seen Figure 2. In isolation, eviction policies influence **speed** aspect of a caching system. We support a couple broad categories of eviction policies (full details is in Appendix B):

• **Temporal policies.** LRU (Least Freqently Used) and LFU (Least Frequently Used) are classic examples that track and select the victim by access-time or access-frequency.

• **Spatial policies.** SB (Score-Based) and FLD (Furthest Layer Distance) select the victim based on its physical characteristic such as gate's logits or distance from current layer.

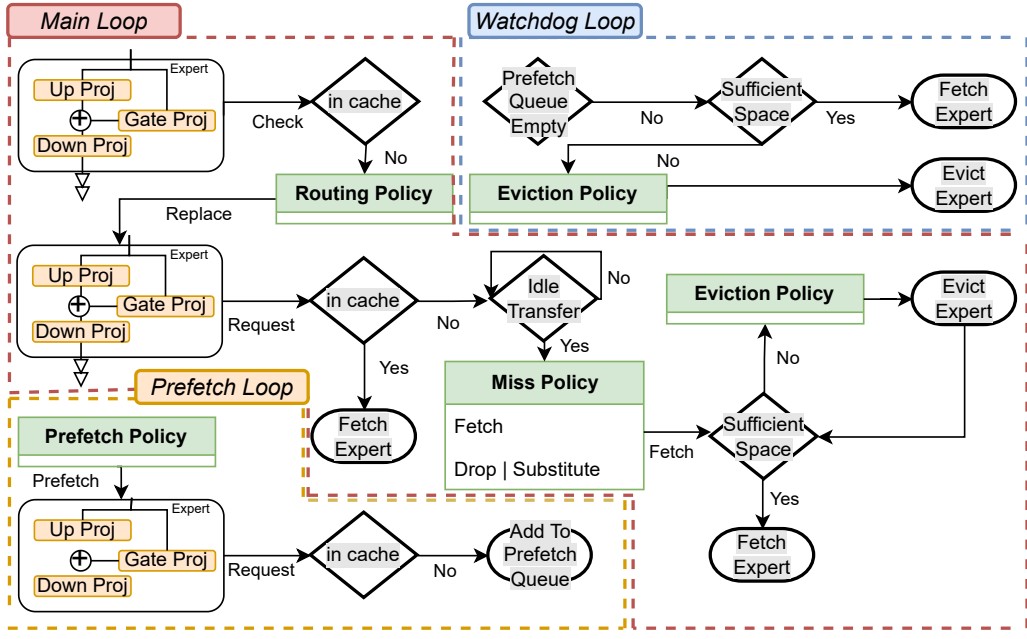

*Figure 2.* Main(red) and Prefetch (yellow) loops operate synchronously and are directly invoked by the model's after gate. Watchdog (blue) operates asynchronously, constatnly monitoring the prefetch queue.

### 3.1.4. PREFETCH STRATEGIES

Prefetch strategies (illustrated in the Prefetch Loop of Figure 2) load experts before they are requested, primarily influencing **speed**. The idea is to utilize the gate from an upcoming layer together with the activations of the current layer to predict future required experts. In our experiments we resort to prefetching only the very next layer experts.

Prefetch operates asynchronously via the Watchdog Loop (shown in blue in Figure 2) that monitors capacity and enqueues prefetch requests to the GPU when space is available.

We support two configuration dimensions in SpecMD:

• **Top-k prefetch with overfetch.** The standard approach prefetches the top-k experts per layer based on router weight magnitude, where k is model-specific ($k = 8$ for OLMoE, $k = 2$ for Mixtral). We introduce an overfetch factor that multiplies the prefetch count to increase bandwidth utilization: with overfetch factor 1.5, OLMoE prefetches 12 experts instead of 8. This trades prefetch precision for cached experts recall at the expense of bandwidth.

• **Score-based approach.** Following Zhu et al. (2025), this strategy filters experts by softmax score rather than fixed top-k count and prefetch all experts above a configurable percentile threshold (the default is 80th percentile). Unlike top-k approaches, this method adapts the number of prefetched experts based on score distribution.

### 3.2. Implementation Overview

To avoid modifying a model's internal implementation, we utilized Pytorch forward hooks to intercepts and modify MoE gate outputs. We also standardize the gate-output protocal interface across architectures (Mixtral, OLMoE, Qwen) for more effective code-reuse. In Algorithm 1 we show the core hook processing across three phases: routing modification, cache management, and prefetch submission. Complete implementation details appear in Appendix A.

## 4. Least-Stale Eviction Policy

In this section, we present our novel policy contribution Least-Stale, which addresses some of the gaps presented in traditional eviction policies for performance improvement.

### 4.1. Motivation

MoE expert access follows a deterministic front-to-back layer sequence (layer $0\rightarrow1\rightarrow2\rightarrow\cdots\rightarrow15$) within each forward pass. Once we move on from a layer, that layer's experts will not be needed until the next forward pass. Temporal-only policies such as LRU and LFU utilize access-time and access-frequency as indicators for expert importance, and thus treat a recently accessed expert as important even though it is now irrelevant to the current forward pass. On the other hand, spatial-only policies like FLD and

**Algorithm 1** Gate Hook Processing

1: **Input:** Gate outputs, layer $l$
2: $(experts, weights, logits, router\_weights) \leftarrow$ Gate outputs
3: // *Phase 1: Routing modification*
4: $logits' \leftarrow$ RoutingPolicy.apply$(l, logits)$
5: $experts' \leftarrow$ top_k(softmax$(logits'))$
6: $weights' \leftarrow router\_weights[experts']$
7: // *Phase 2: Cache management (during expert execution)*
8: **for** each expert $e$ in $experts'$ **do**
9:     **if** $e \notin$ GPU_Cache **then**
10:         MissPolicy.handle$(e)$
11:     **end if**
12: **end for**
13: // *Phase 3: Prefetch submission*
14: **if** $l$ has prefetch targets $\mathcal{T}$ **then**
15:     **for** each target layer $t \in \mathcal{T}$ **do**
16:         $predicted \leftarrow$ PrefetchPolicy.predict$(t)$
17:         PrefetchQueue.submit$(predicted, t)$
18:     **end for**
19: **end if**
20: **Return:** $(experts', weights', logits', router\_weights)$

Score-Based use layer distance or router scores to indicate importance with no understanding of temporal relevancy. Regardless of approach, the outcome is the same: experts from future layers get evicted prematurely. We show this in Figure 3-top where the immediate next layer experts are most likely to be evicted. This observation motivates our Least-Stale approach, which exploits the spatial-temporal structure of expert access to prevent premature evictions.

### 4.2. The Least-Stale Policy

Eviction policies determine which cached experts to evict when capacity is exceeded. In many instances, eviction policies can cause **collision misses**, a type of cache miss where experts are evicted and then immediately needed again in the same forward pass. To address collision misses, Least-Stale (LS) combines both temporal factors (access time) and spatial awareness (layer positioning) to minimize collision misses.

*Staleness and position.* An expert is considered *stale* if it was accessed in a previous forward cycle; otherwise, it is *current*, meaning it is used in the current pass or selected by prefetching. We describe an expert as *left* if it appears earlier than the current execution point in layer–expert order.

Figure 3-bottom illustrates how staleness and position jointly determine eviction priority. The cache is partitioned into a *stale queue* containing experts from previous forward passes and a *current queue* containing experts from the

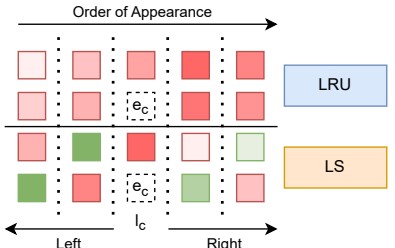

*Figure 3.* **Top:** LRU eviction as a purely temporal policy. Blocks with higher intensity indicate higher eviction priority; consequently, experts from the immediately following layer are most likely to be evicted. **Bottom:** Illustration of the Least-Stale (LS) policy, which combines temporal and positional signals. Cached experts are marked as current (green, accessed in the current forward pass) or stale (red, accessed in a previous pass), while layer index encodes position. Higher color intensity corresponds to higher eviction probability. Stale experts have a higher eviction priority than current ones.

ongoing pass.

Eviction prioritizes stale experts, as recently used experts are more likely to be reused by upcoming tokens or refreshed by prefetching. Experts in the current queue are evicted only to prevent out-of-memory errors. Within each queue, experts follow FIFO ordering based on layer position, favoring spatial and temporal locality.

In practice, both queues are implemented as priority heaps, requiring $\mathcal{O}(N)$ space, $\mathcal{O}(\log N)$ insertion, and $\mathcal{O}(1)$ eviction, where $N$ is the cache capacity.

## 5. Evaluation and Results

Having motivated and described Least-Stale, we now present comprehensive evaluation results that validate our design choices and characterize performance across the full policy space.

### 5.1. Experimental Setup

To effectively traverse the huge policy decision-space across various models and capacity constraints, we adopt a staged evaluation strategy: we first isolate optimal configuration from performance-neutral policies, i.e eviction policies and prefetch policies, then evaluate quality-speed trade-offs using optimal configurations from early stages. This approach reduces the search space while ensuring fair comparison.

• **Models.** We evaluate across four MoE architectures: OLMoE-1B-7B (Muennighoff et al., 2025), Mixtral-8x7B (Jiang et al., 2024), Qwen1.5-MoE-A2.7B(Team, 2024), and Phi-3.5-MoE (Abdin et al., 2024). Our selection covers both wide MoE with many small experts and deep MoE with a few yet large experts, enabling us to characterize how policy effectiveness varies with a model structure.

● **Tasks.** To measure realistic caching behaviour (with both prefill and decoding stages) we test tasks using autoregressive generation. These tasks are: GSM8K (Cobbe et al., 2021) (mathematical reasoning), TruthfulQA (Lin et al., 2021) (factuality), and NaturalQuestions(Kwiatkowski et al., 2019) (question answering). Performance is measured via exact match for GSM8K and GPT-5 as judge for QA tasks. To make the policy sweep in Sections 5.2–5.4 tractable, we evaluate each policy on a 100-example subsample per task; the final mix-policy comparison in Table 1 uses the full task dataset minus those 100 sweep examples.

● **Hardware constraints.** We evaluate all models on a single A100 GPU with 80GB VRAM. We software-limit our GPU cache to 1%, 5%, or 25% of full model size, which amounts for example, to at most 21GB and as few as 0.8GB for Mixtral. We opted to use device native bandwidth, which on our device measures on average 5 GB/s. We quantize experts into int8 and int4 using bitsandbytes (Dettmers et al., 2022) and optionally int2 using HQQ (Badri & Shaji, 2023)to evaluate cache miss policy comprehensively. For detailed model specifications please see in Appendix E.

### 5.2. Least-Stale Effectiveness

We first validate the Least-Stale policy by analyzing its impact on collision misses and inference speed across different models and capacity levels. We can see how baseline eviction policies at various capacities scored higher collision rates in Figure 4 which presents a 5% cache capacity. More extensive results can be found in Figure 8 of the Appendinx.

For 5% cache capacity in particular, Least-Stale achieves 1.6-1.9% collision rates, while LRU reaches 4.5-12.6% and SB suffers catastrophic 42.6-60.9% rates. SB performs worst because historical gate scores capture neither spatial position nor temporal relevance; a highly-scored expert from layer 2 gets retained over a lower-scored expert from layer 10, even though layer 10 is spatially next. FLD, unlike LRU, tracks only expert distance from current layer and performed only slightly better than LRU.

● **Spatial–Temporal awareness.** Least-Stale determines eviction priority by forward-pass cycles. Experts accessed in the *current* pass are protected to prevent premature eviction of prefetched future experts and to increase the odds of reuse in the next forward cycle for left experts. Experts from *previous* passes are marked as stale and prioritized for eviction, unless they are needed again via prefetching.

In Figure 5, we illustrate the per-layer collision misses. Baseline policies tend to accumulate throughout the forward pass as they evict soon-needed experts. At 1% capacity, LRU and Score-Based reach 8-12% collision rates in deeper layers. Least-Stale maintains near-zero collisions across all layers by only evicting stale left-layer experts.

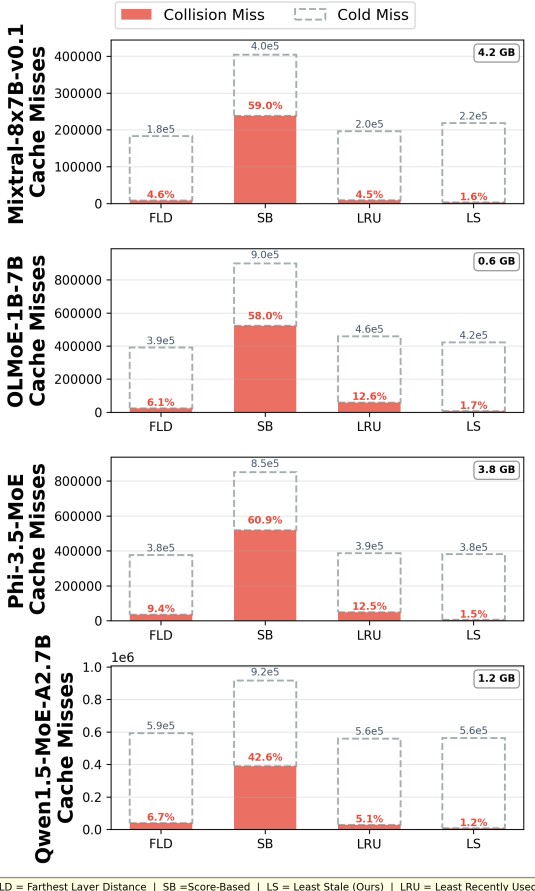

*Figure 4.* Collision miss across four eviction policies (FLD, SB, LRU, LS) at 5% capacity threshold showing total misses (gray) and collision misses (red). Refer to Section 3.1.3 for abbreviations.

● **Speed Impact.** The collision reduction translates directly to TTFT gains. At 5% capacity, Least-Stale reduces collision misses by 2.6-8.6× versus LRU and 58-91× versus Score-Based. When used as a drop-in replacement for existing approaches' eviction policies (keeping their prefetch and routing unchanged), Least-Stale achieves 10.7-34.7% TTFT improvements on OLMoE (Figure 1), with consistent gains of 14.8-44.1% across diverse baselines systems (Table 3).

### 5.3. Prefetch Strategy Effectiveness

> **Key Insight: Prediction accuracy $\neq$ cache performance**
>
> Score-based prefetching outperforms top-k despite lower overall prediction accuracy. Giving each layer the ability to adapt the number of experts to prefetch based on gating scores proves more effective at utilizing available bandwidth that a fix threshold.

● **Understanding the mismatch.** Using Least-Stale as our baseline eviction policy, we compare two prefetch strategies: top-k prefetching with different multipler rates and

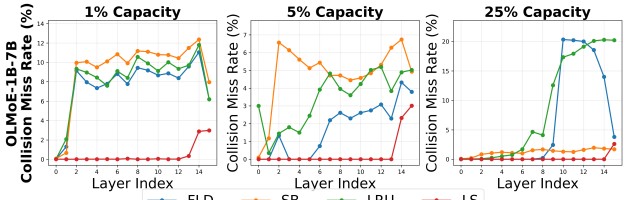

*Figure 5.* Per-layer collision miss rates for OLMoE across 16 layers. Least-Stale (red) maintains near-zero collisions across layers.

score-based percentile filtering. What we discover is that higher prediction accuracy does not guarantee better cache performance in term of speed.

In Figure 6 we show that top-k with a 1.0 factor achieves high precision and recall across all models. Yet the right column shows that score-based prefetching typically delivers higher hit rates with lower synchronous overhead (time spent waiting on cache misses) for the majority of the model we tested, with the exception of Mixtral.

• **Effective Bandwidth Utilization.** We find that the score-based approach more effectively adapts the number of experts to prefetch on a per-layer basis. Simply scaling up a fixed number of experts does not match the same performance. We conclude that adaptive prefetching more effectively allocates existing bandwidth to high-reward layers while reducing the number of experts prefetched for high-risk layers, thereby outperforming fixed top-k strategies in most cases.

• **The Exception.** We observe Mixtral as the exception. It is a deep MoE with a small total number of experts, but each expert is large (∼336MB). For these types of MoEs, the simpler top-k strategy performs competitively. This occurs because the size of the experts is too large for the available bandwidth, which limits the benefit of a dynamic prefetch system. This suggests that deep MoEs with large expert sizes may be better suited to simpler top-k prefetching.

### 5.4. Quality-Speed Trade-offs

• **Trading Quality for Speed.** With optimal eviction and prefetch policies established, we now examine how different miss handling and routing strategies perform. By modifying these two policies, we can achieve a quality/speed tradeoff, shown in Figure 7.

when studying miss-handling policies, we find that Fetch Priority (blue squares) consistently occupies the Pareto frontier across models. This approach stores multiple precision levels and selectively degrades low-importance experts (bottom 60th percentile) during cache misses. For Mixtral at 8-bit, we observe 10%-12% performance improvements with 50-60% speed gains. For OLMoE at 4-bit, Fetch Priority maintains near-baseline performance (0-5% loss) while achieving 20%-30% speedup.

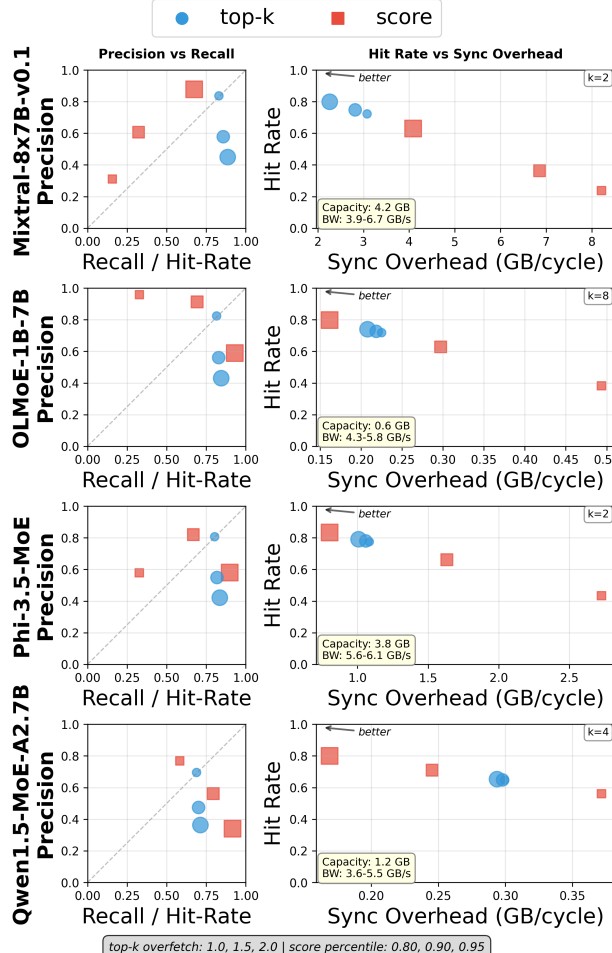

*Figure 6.* Prefetch strategy comparison at 5% capacity. Left: Precision vs recall shows top-k 1.0× achieves highest prediction accuracy. Right: Hit rate vs synchronous overhead reveals score-based 80th percentile achieves better cache performance for most models through dynamic adaptation.

• **The Trade-offs.** Drop Priority, as expected, provides the highest speed improvements at the cost of performance degradation. Qwen at 4-bit suffers -25% to -30% performance loss, while OLMoE, a wide MoE, shows only -5% to -15% degradation. This effectively demonstrates that more active experts can dilute the impact of dropping any single expert. We find that Fetch Lowest Bit and Drop Priority Cascade show inconsistent results across models, offering no clear advantage over simpler strategies. On the other hand, Substitution Score fails across all configurations, suggesting that replacing missing experts with functionally similar alternatives is unreliable, regardless of capacity level.

• **Cache-Aware Routing Impact.** We find that cache-aware routing is credited for 10%-20% of the achieved speed improvements in some cases (compare different intensities of the blue squares on each figure). The effectiveness depends critically on model architecture: OLMoE can tolerate higher routing bias ($\lambda$) than Mixtral, since wider expert distribu-

*Table 1.* Policy comparison across all models at 5% cache capacity (4bit quantization, 5 GB/s bandwidth, with QKV Cache). TTFT in milliseconds. Each configuration represents a distinct point in the quality-speed trade-off space identified through SpecMD's policy evaluation (Section 5.2 and Figure 7). No single configuration dominates across all models.

| Config | Policy Prefetch | Evict | Miss | Route | OLMoE-1B-7B GSM8K | Truth. | NQ | TTFT | Token/s | Mixtral-8x7B GSM8K | Truth. | NQ | TTFT | Token/s |
|---|---|---|---|---|---|---|---|---|---|---|---|---|---|---|
| | | | | Baseline (8bit) | 0.704 | 0.540 | 0.361 | 3166 | 1.5 | 0.569 | 0.345 | 0.510 | 10296 | 0.4 |
| #1 | Spec | LRU | Fetch | Stnd | 0.677 | 0.555 | **0.343** | 3077 | 1.91 | **0.566** | 0.343 | **0.518** | 4238 | 0.98 |
| #2 | Score | ScoreE | Subs | Stnd | 0.073 | 0.493 | 0.159 | 2257 | 1.22 | 0.037 | 0.251 | 0.273 | **1620** | 0.75 |
| #3 | Spec | LHU | Cascade | Stnd | 0.606 | 0.548 | 0.291 | **1590** | 1.34 | 0.310 | 0.326 | 0.347 | 2322 | 0.71 |
| #4 | None | LRU | Fetch | Mod | 0.664 | 0.558 | 0.332 | 2665 | 2.33 | **0.566** | 0.326 | 0.517 | 4333 | 1.07 |
| #5 | Score | LS | Fetch | Stnd | **0.691** | **0.559** | 0.331 | 2220 | 1.69 | 0.528 | **0.356** | 0.471 | 3044 | 1.42 |

| Config | Policy Prefetch | Evict | Miss | Route | Phi-3.5-MoE GSM8K | Truth. | NQ | TTFT | Token/s | Qwen1.5-MoE-A2.7B GSM8K | Truth. | NQ | TTFT | Token/s |
|---|---|---|---|---|---|---|---|---|---|---|---|---|---|---|
| | | | | Baseline (8bit) | 0.697 | 0.774 | 0.594 | 7902 | 0.8 | 0.471 | 0.595 | 0.460 | 5728 | 2.3 |
| #1 | Spec | LRU | Fetch | Stnd | **0.705** | 0.775 | 0.589 | 4452 | 1.69 | **0.506** | **0.614** | 0.473 | 3724 | 2.89 |
| #2 | Score | ScoreE | Subs | Stnd | 0.027 | 0.514 | 0.269 | **1906** | 0.86 | 0.008 | 0.480 | 0.408 | 2739 | 1.11 |
| #3 | Spec | LHU | Cascade | Stnd | 0.604 | 0.743 | 0.563 | 3459 | 1.05 | 0.217 | 0.608 | 0.462 | 2492 | 2.19 |
| #4 | None | LRU | Fetch | Mod | **0.705** | 0.777 | **0.591** | 5025 | 1.72 | 0.296 | 0.611 | 0.465 | 3489 | 3.50 |
| #5 | Score | LS | Fetch | Stnd | 0.684 | **0.780** | 0.587 | 3863 | 1.11 | 0.497 | 0.607 | **0.474** | 2893 | 2.11 |

*Prefetch:* Spec (speculative) (Eliseev & Mazur, 2023), Score (score-based) (Zhu et al., 2025), None. *Evict:* LRU (Eliseev & Mazur, 2023), LHU (Tang et al., 2024), ScoreE (Zhu et al., 2025), LS (Least-Stale, ours). *Miss:* Fetch (blocking load), Cascade (precision) (Tang et al., 2024), Subs (substitution) (Zhu et al., 2025). *Route:* Stnd (standard), Mod (Skliar et al., 2025).

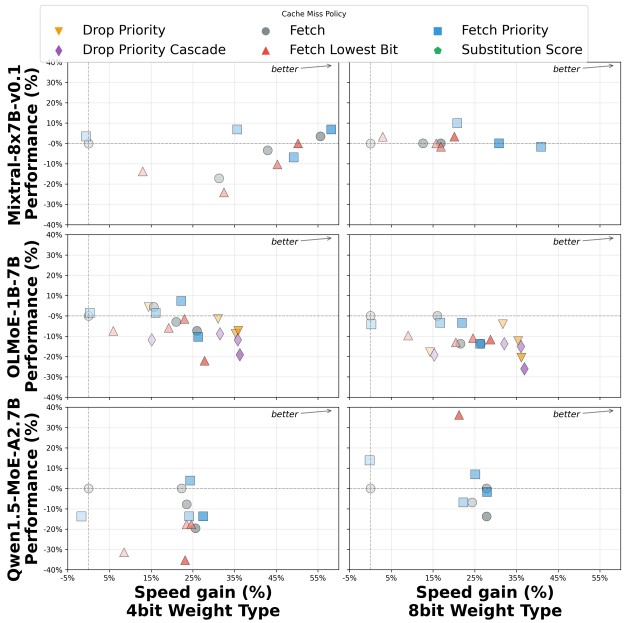

*Figure 7.* Quality-memory trade-offs for six miss handling policies at 5% capacity. Performance and speed shown relative to Fetch policy baseline. We report the average accuracy among the tasks under study as performance. Points above the zero horizontal line indicate performance improvements compared to the baseline; points to the right of the zero vertical line indicate speed gains. Color intensity indicates cache-aware routing strength ($\lambda$). Substitution policies fall outside the chart range.

tions can redistribute the impact of routing perturbations more effectively. Our experiments show that $\lambda$ values between 0.1-0.5 work best, balancing speed gains with quality preservation; too conservative values provide minimal benefit, too aggressive values degrade quality unacceptably.

### 5.5. SpecMD Mix-Policy Evaluation

SpecMD's enables an easy way to mix and run ad-hoc policies across all four dimensions (eviction, prefetch, miss handling, routing) on the same testing platform. Through it, we can identify policies with favorable characteristic via policy evaluation. In previous sub-sections, we discussed at length policy interactions and the various tradeoffs; we now selected five set of mix policies to compare running full task evaluation. This we summarize in Table 1 using 5% capacity cache capacity and 4bit quantization.

• **Configuration selection methodology.** To ensure fair comparison and avoid cherry-picking, we select five configurations from our policy evaluation (Section 5.3) that span the quality-speed trade-off space with sufficient policy diversity. Configurations #1–#4 represent existing approaches from prior work, while Config#5 combines our proposed Least-Stale eviction with score-based prefetching.

• **Results interpretation.** Configurations selected using SpecMD perform well in most cases, but this is not the point. The point of SpecMD is the experimental platform which enables flexible configuration and unbiased study of the policies and their interactions. Using SpecMD, re-

searchers can select optimal policy combinations based on their specific hardware constraints and quality requirements and share their findings more effectively.

• **Batch-size scaling.** To check that the policy ranking does not collapse using more batch size of more than one, we re-run the five configurations on OLMoE-1B-7B (GSM8K, full dataset, 5% cache capacity) at batch sizes 4 and 8 (Table 2). The qualitative ordering is preserved at every batch size: Config #1, #4, and #5 stay within noise distribution, Config #2 collapses, and Config #3 degrades sharply.

*Table 2.* Batch-size scaling on OLMoE-1B-7B / GSM8K at 5% cache capacity. Policy rankings are preserved; Configurations #1, #4, and #5 remain top-tier across all batch sizes.

| Config | Batch=1 | Batch=4 | Batch=8 |
|--------|---------|---------|---------|
| #1 | 0.677 | 0.660 | 0.655 |
| #2 | 0.080 | 0.048 | 0.024 |
| #3 | 0.592 | 0.390 | 0.379 |
| #4 | 0.664 | 0.663 | 0.657 |
| #5 | 0.675 | 0.657 | 0.637 |

## 6. Limitations

SpecMD considers throughput and capacity as the two primary hardware constraints. Real deployments may face additional challenges—DMA overhead, memory-bus contention, thermal throttling—that we do not model. We believe our observations generalize across these settings, but cannot account for every deployment-specific circumstance. Our experiments also focus on single-GPU inference. Multi-GPU and pipeline-parallel settings introduce expert placement and cross-device transfer dynamics that interact with caching in non-trivial ways, and form an interesting next step. Finally, we limit our scope to training-free, drop-in policies and do not consider training-based approaches such as expert pruning or routing-aware fine-tuning, which target a different region of the cost-quality frontier.

## 7. Conclusion

This work presented SpecMD, a unified benchmark for evaluating MoE expert caching policies. Through controlled hardware constraints and minimal PyTorch integration, we enabled reproducible comparison of caching strategies without specialized hardware requirements.

Our study challenges traditional caching assumptions for MoE architectures. We demonstrated that expert access follows deterministic sequential layer patterns rather than temporal locality, rendering conventional LRU/LFU approaches ineffective. The proposed Least-Stale eviction policy exploits this structure, achieving 3-9× collision miss reduction at practical capacity levels, translating as an average 17% improvement in TTFT, across different models and meth-

ods. Counterintuitively, we revealed that prediction accuracy does not guarantee cache performance—score-based prefetching delivers superior hit rates through dynamic bandwidth adaptation despite lower precision metrics.

## Impact Statement

This paper presents work whose goal is to advance the field of Machine Learning. There are many potential societal consequences of our work, none which we feel must be specifically highlighted here.

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

# A. Implementation Details

## A.1. Gate Hook Architecture

Our framework intercepts MoE gate layer outputs through PyTorch forward hooks. When a gate layer executes, it outputs a 4-tuple: selected expert indices, routing weights for those experts, raw router logits, and normalized router weights across all experts. We wrap gate outputs to ensure consistent formatting across different MoE architectures (Mixtral, OLMoE, Qwen), providing a unified interface for downstream components. This standardization eliminates model-specific handling in policy implementations.

The hook manager registers forward hooks on each gate layer to intercept these outputs. Algorithm 1 (main text) outlines the core hook logic. Upon receiving gate outputs, the routing policy can modify logits to prefer GPU-resident experts. When routing modifies expert selections, we apply dual-logit semantics: modified logits determine which experts to select, but original logits determine their contribution weights. This preserves model output integrity—even if cache-aware routing changes which experts execute, their contributions are weighted according to the original router decisions.

## A.2. Cache Manager Details

For each selected expert during model execution, the cache manager checks GPU cache availability. Cache hits proceed immediately. Cache misses trigger the miss handling policy: fetch blocks until the expert loads from CPU, drop skips low-rank experts, or substitution replaces missing experts with cached alternatives. The cache manager maintains experts across CPU (pinned memory) and GPU (active cache) storage, supporting mixed-precision configurations where experts exist in multiple quantization levels (fp16, int8, int4, int2).

## A.3. Prefetch Manager and Watchdog Thread

Concurrently, the prefetch manager anticipates future expert needs. The hook manager uses either current gate outputs or future gate functions on current gate logits to predict which experts subsequent layers will require, submitting prefetch requests to an asynchronous queue. A watchdog thread monitors this queue and GPU capacity, issuing non-blocking CUDA transfers when space allows. Prefetched experts populate the GPU cache before they are requested, reducing miss frequency. When capacity pressure exceeds limits, the eviction policy identifies victims—our Least-Stale policy prioritizes removing experts from already-processed layers while protecting experts needed for upcoming layers.

### A.4. Hardware Emulation

Hardware constraints are emulated through software limits rather than requiring physical low-bandwidth devices. Bandwidth throttling delays expert transfers between CPU and GPU to simulate slower interconnects. Capacity limits restrict the available GPU cache below physical VRAM, forcing the system to make eviction and prefetch trade-offs. This emulation enables systematic exploration: evaluating a policy under 5 GB/s bandwidth and 25% capacity requires only configuration changes, not specialized hardware.

## B. Baseline Eviction Policies

We implement five baseline eviction strategies for comparison:

**LRU (Least Recently Used)** evicts the least recently accessed expert. However, MoE expert access follows deterministic sequential layer patterns rather than temporal patterns, making LRU ineffective for expert caching.

**LFU (Least Frequently Used)** evicts the expert with the lowest access count over the entire inference history. This policy prioritizes retaining frequently-accessed experts across all layers. However, it fails to account for layer-wise access patterns where experts from completed layers will not be needed until the next forward pass.

**LHU (Least High-precision Used)** (Tang et al., 2024) tracks high-precision expert usage frequency, prioritizing retention of frequently-accessed high-precision experts since cache misses for these incur significantly higher latency. This policy must be combined with fetch cascade precision policies (e.g., int8→int4→int2) that allow graceful degradation to lower precision on cache misses.

**FLD (Farthest Layer Distance)** (Tang et al., 2024) exploits layer-wise structure by retaining experts from nearer layers, which are more likely to be needed soon. This policy computes the distance between each cached expert's layer and the current inference layer, evicting experts from the farthest layers first.

**Score-based** (Zhu et al., 2025) evicts experts with the lowest historical gate scores, using router activation values as importance signals. This policy maintains a running history of gate scores for each expert and evicts those with minimal contribution to model outputs. However, historical scores do not capture position in the forward pass, causing premature eviction of soon-needed experts.

## C. Complete Eviction Policy Results

Figure 8 presents complete collision miss rate results referenced in Section 5.1, showing consistent performance across all capacity levels.

Figure 9 shows per-layer collision patterns for all models, revealing the mechanism behind Least-Stale's effectiveness across different architectures.

## D. Least-Stale Eviction as Drop-in Improvement

Beyond our full system, we evaluate the Least-Stale eviction policy as a standalone contribution that can improve existing approaches. As visualized in Figure 1 for OLMoE-1B-7B, Least-Stale provides consistent speedups across diverse baseline systems. Table 3 shows comprehensive TTFT improvements across all four models when we replace each baseline's original eviction policy with Least-Stale while keeping their other components (prefetching, routing) unchanged.

Least-Stale eviction provides consistent TTFT improvements across most configurations (10-44% reduction), demonstrating its value as a portable component. However, SpecMD's advantage stems from the synergistic combination of eviction, prefetching, and cache-aware routing policies.

## E. Model Specifications

Table 4 provides detailed specifications for all models evaluated in this work.

These specifications highlight the diversity of architectural choices: OLMoE uses many small experts with high top-k (fine-grained sparsity), while Mixtral uses few large experts with low top-k (coarse-grained sparsity). This diversity enables us to characterize policy effectiveness across different granularity regimes.

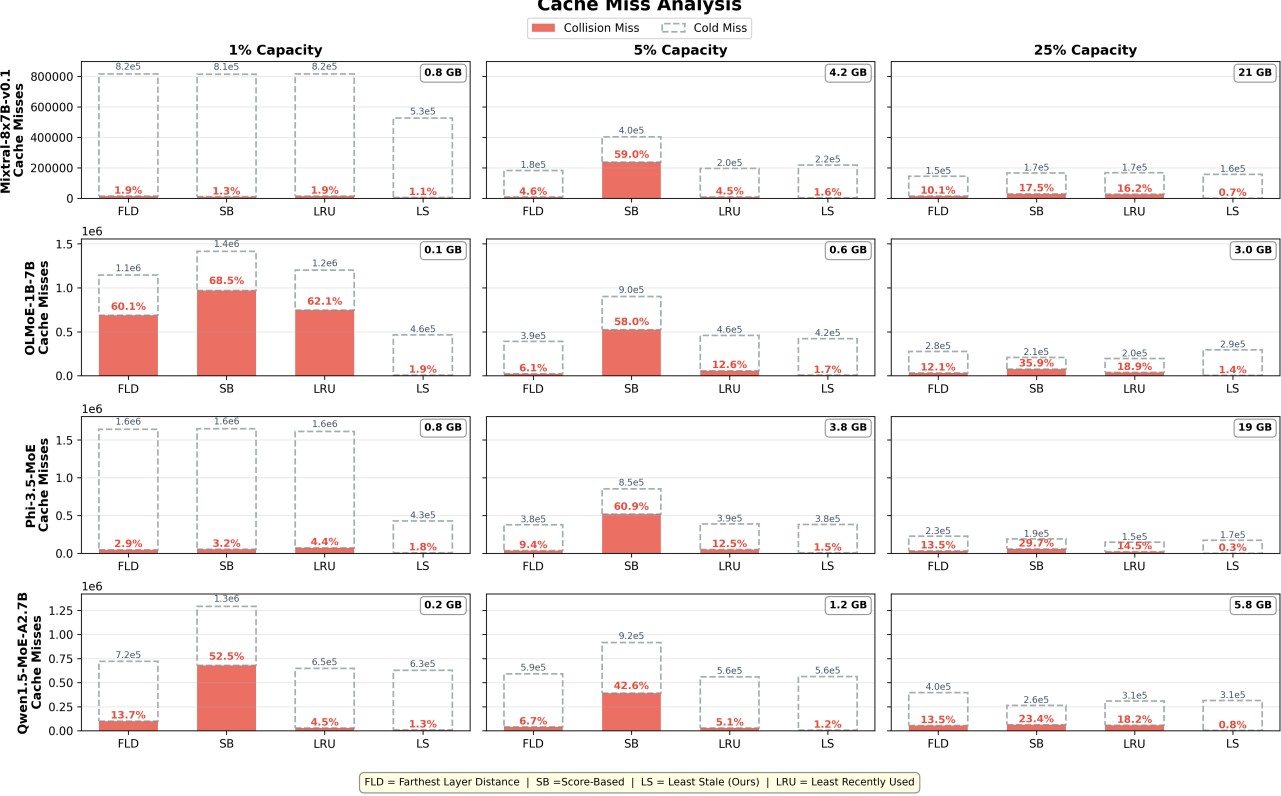

*Figure 8.* Complete collision miss analysis across four eviction policies (FLD, SB, LRU, LS) at three capacity levels (1%, 5%, 25%). Least-Stale (LS) achieves dramatically lower collision miss rates (red bars) compared to baseline policies across all models and capacity constraints. The pattern is consistent: LS achieves 10-34× lower collision rates than baseline policies.

| Model | Approach | Original TTFT (ms) | w/ Least-Stale (ms) | Improvement (%) |
|-------|----------|-------------------|--------------------|-----------------|
| *OLMoE-1B-7B* | | | | |
| | Eliseev '23 | 3076.8 | 2748.1 | 10.7% |
| | Zhu et al. '25 | 2257.2 | 1473.1 | 34.7% |
| | HOBBIT '24 | 1590.1 | 1309.2 | 17.7% |
| | MixtureCache '25 | 2664.6 | 2297.5 | 13.8% |
| *Mixtral-8x7B* | | | | |
| | Eliseev '23 | 4238.0 | 3485.2 | 17.8% |
| | Zhu et al. '25 | 1620.3 | 1460.8 | 9.8% |
| | HOBBIT '24 | 2321.7 | 1725.2 | 25.7% |
| | MixtureCache '25 | 4332.6 | 3573.2 | 17.5% |
| *Phi-3.5-MoE* | | | | |
| | Eliseev '23 | 4451.7 | 3791.1 | 14.8% |
| | Zhu et al. '25 | 1905.6 | 1987.6 | -4.3% |
| | HOBBIT '24 | 3458.6 | 1933.6 | 44.1% |
| | MixtureCache '25 | 5025.3 | 3791.1 | 24.6% |
| *Qwen1.5-MoE-A2.7B* | | | | |
| | Eliseev '23 | 3724.0 | 3433.4 | 7.8% |
| | Zhu et al. '25 | 2739.2 | 2620.9 | 4.3% |
| | HOBBIT '24 | 2492.0 | 2033.7 | 18.4% |
| | MixtureCache '25 | 3488.6 | 2685.4 | 23.0% |

*Table 3.* Least-Stale eviction policy as drop-in improvement for existing approaches. We replace each baseline's original eviction policy with Least-Stale while keeping their prefetching and routing strategies unchanged. Positive percentages indicate TTFT reduction (lower is better). This demonstrates Least-Stale's generalizability across different system architectures.

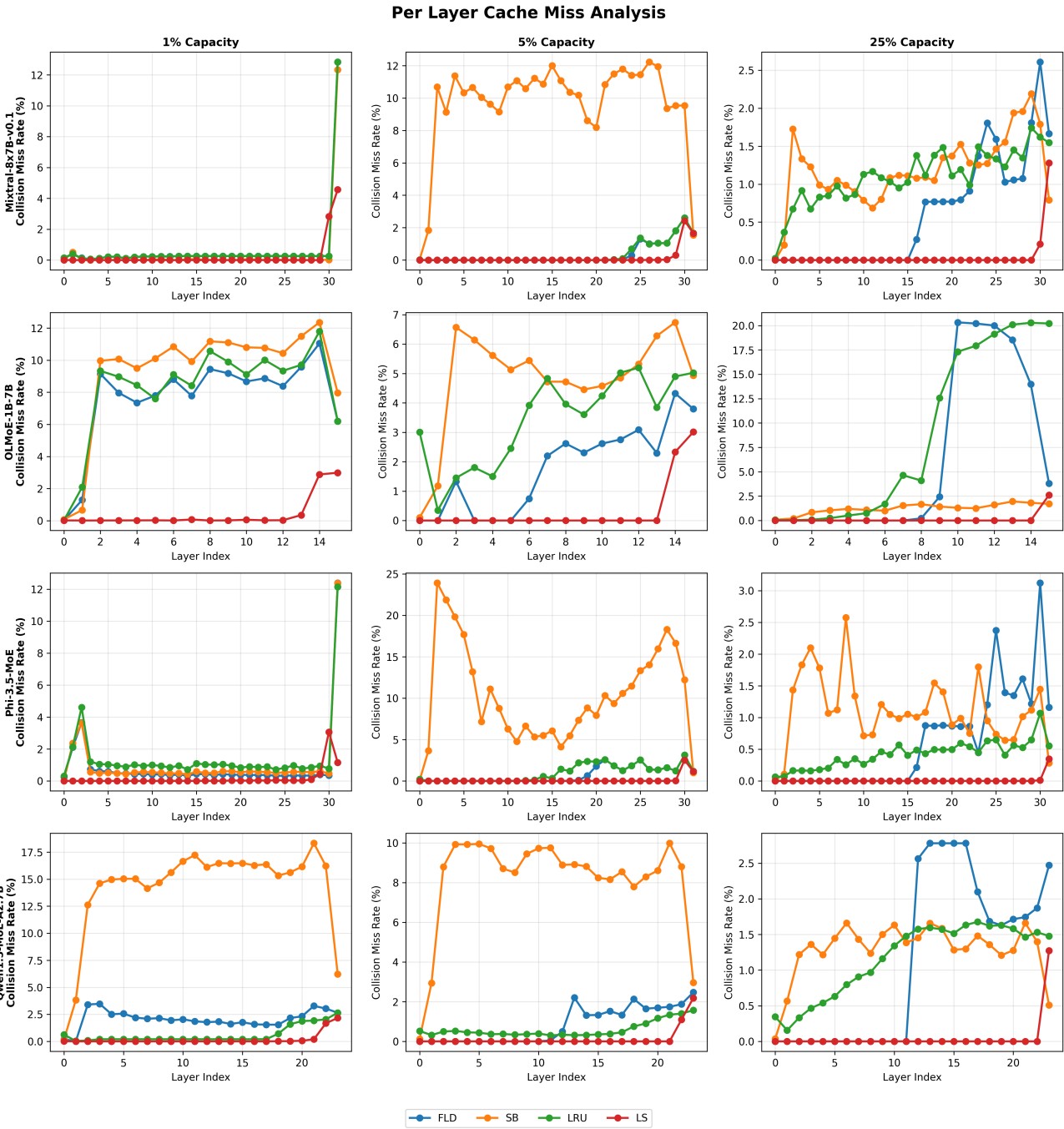

*Figure 9.* Per-layer collision miss analysis across all four models at three capacity levels (1%, 5%, 25%). Least-Stale (red) maintains near-zero collisions by protecting experts in the current forward pass, while baseline policies show accumulating collisions throughout inference. The mechanism reveals why Least-Stale achieves dramatic improvements: it only evicts stale left-layer experts while baseline policies prematurely evict soon-needed experts.

*Table 4.* Detailed specifications for evaluated MoE architectures. All sizes in GB unless specified. Activated size represents memory required per forward pass.

| Model | Layers | Experts per Layer | Top-k | Expert Size | Total Size | Expert Params | Non-Expert | Activated |
|---|---|---|---|---|---|---|---|---|
| OLMoE-1B-7B | 16 | 64 | 8 | 12 MB | 12.89 GB | 12.0 GB | 0.89 GB | 2.39 GB |
| Mixtral-8x7B | 32 | 8 | 2 | 336 MB | 86.99 GB | 84.0 GB | 2.99 GB | 23.99 GB |
| Qwen1.5-MoE-A2.7B | 24 | 60 | 4 | 16.5 MB | 26.67 GB | 23.2 GB | 3.46 GB | 5.01 GB |
| Phi-3.5-MoE | 32 | 16 | 2 | 150 MB | 77.99 GB | 75.0 GB | 2.99 GB | 12.37 GB |

