# OpenReview forum: "SpecMD: A Comprehensive Study On Speculative Expert Prefetching"
_ICML.cc/2026/Conference — ICML 2026 regular_

### Official Review · Reviewer_jD99 · 2026-03-09

**Soundness:** 3
**Presentation:** 2
**Significance:** 3
**Originality:** 3
**Overall Recommendation:** 4
**Confidence:** 4

**Summary:**

This work introduces a standardized benchmarking framework (SpecMD) for MoE deployment and an effective eviction policy, Least-Stale. Experiments across multiple MoE models, hardware constraints, and policy combinations addresses a notable gap in prior work. The careful experimental design and insightful analysis of policy interactions provide valuable contributions to MoE system research.

**Compliance With Llm Reviewing Policy:**

Affirmed.

**Key Questions For Authors:**

see weaknesses

**Limitations:**

It is necessary to discuss whether the experimental findings obtained on a single GPU can generalize to more complex deployment scenarios.

**Strengths And Weaknesses:**

S1. This benchmark study provides a comprehensive comparison of a wide range of approaches across multiple dimensions.

S2. The proposed Least-Stale eviction policy demonstrates strong empirical performance while remaining simple and practical to implement.

W1. The experimental setting is somewhat specialized. The paper relies on software emulation of hardware constraints on a single A100 80GB GPU. In practice, large-scale industrial MoE models (e.g., Mixtral-8x7B, DeepSeek-V2) are typically deployed using multi-GPU distributed inference, where experts are sharded across different GPUs. The current study focuses exclusively on single-GPU inference and does not analyze how the proposed Least-Stale policy would adapt to distributed environments,

W2. A major practical challenge in MoE cache management arises in long-sequence autoregressive inference, where cache pressure accumulates continuously and the impact of eviction strategies becomes more pronounced. It would be better for the authors to provide a systematic analysis of policy performance across varying sequence lengths.

W3. The eviction, prefetching, and miss-handling strategies included in the comparison are mostly classical approaches from the academic literature. The evaluation does not cover several mainstream cache optimization techniques widely adopted in industrial MoE inference systems. Including such baselines would strengthen the practical relevance of the study.

---

> ### Author Rebuttal · Authors · 2026-03-30
>
> We thank the reviewer for their support.
>
> **W1 — Single-GPU, no multi-GPU distributed inference**
>
> Thank you for the chance to clarify.
> We agree that multi-GPU support would be a useful perspective, but its complex distribution nature would make analyzing the interaction among various MoE polices difficult. Hence, in SpecMD, we focused on a single-GPU setup where we can easily contrast the benefits of different polices easy. Also, while not always true, there is a high chance that we could afford expert sharding over multiple-GPUs (a popular MoE distribution stratgy) which would require small or no expert caching by nature. We will make this scope explicit in the introduction.
>
> **W2 — Varying sequence length analysis**
>
> We thank the reviewer for this suggestion, should the paper be accepted, we will add a additional figure showing TTFT and collision miss rate as a function of input sequence length (128, 512, 2048 tokens).
>
>
> **W3 — Missing industrial baselines**
> We thank the reviewer for this suggestion. We welcome specific citations to help us identify which industrial techniques to add. Our current comparison covers all published academic baselines we are aware of (Eliseev, HOBBIT, Zhu et al., MixtureCache). To our knowledge, the most popular serving engines for LLMs (vLLM, SGLang) do not currently support MoE expert caching.

---

> > ### Author Rebuttal · Reviewer_jD99 · 2026-04-05
> >
> > Thanks for rebuttal. I hope you will add supplementary experiments in the final version.

---

### Official Review · Reviewer_YGdn · 2026-03-10

**Soundness:** 2
**Presentation:** 3
**Significance:** 3
**Originality:** 2
**Overall Recommendation:** 4
**Confidence:** 4

**Summary:**

The paper tackles the high memory requirements of deploying MoE models on memory-constrained devices by focusing on expert caching mechanisms. Overall, the paper investigates a pertinent question: how different caching policies—namely routing, prefetching, eviction, and miss handling—interact with each other under various hardware constraints. This paper presents SpecMD, a software-emulated benchmarking framework designed to evaluate MoE caching strategies. Using this framework, the authors conduct extensive empirical studies and reveal that standard temporal eviction policies (like LRU and LFU) perform poorly because MoE expert access follows a deterministic, sequential layer pattern rather than temporal locality. Motivated by this, the authors propose Least-Stale, a novel eviction policy that combines temporal and spatial (layer-position) awareness. The experiments demonstrate that Least-Stale significantly reduces collision misses and improves Time-to-First-Token (TTFT) at highly constrained cache capacities.

**Compliance With Llm Reviewing Policy:**

Affirmed.

**Final Justification:**

My concerns have been adequately addressed during rebuttal. Therefore, I will maintain my original score of a weak accept.

**Key Questions For Authors:**

1. How does the proposed methods handle concurrent requests in a dynamic batched inference setting?
2. Are there plans to extend the SpecMD framework to support or benchmark against more recent SOTA expert prefetching methods and system-level expert offloading frameworks?
3. Could you clarify the specific experimental configurations (e.g., sequence length, batch size, prefill vs. decoding phase)?

**Limitations:**

yes

**Strengths And Weaknesses:**

Strengths:
1. The presented SpecMD framework provides a standardized and flexible testbed that facilitates the evaluation and integration of various MoE caching policies across diverse hardware constraints and deployment scenarios.
2. The paper offers valuable empirical insights into the design of expert caching mechanisms.
3. The Least-Stale eviction policy is simple yet effective. By exploiting the spatial-temporal access patterns of MoE layers, it directly addresses and significantly mitigates the issue of collision misses.

Weaknesses:
1. The evaluated prefetching strategies appear restricted in scope, especially given the extensive prior exploration in this domain. Furthermore, the manuscript lacks a comprehensive discussion on the system-level implementation and optimization of expert offloading. Contextualizing the proposed framework with recent advancements in MoE offloading systems (e.g., https://arxiv.org/abs/2411.11217, https://arxiv.org/abs/2502.06888, https://arxiv.org/abs/2503.09716, https://arxiv.org/abs/2308.12066) would significantly strengthen the paper's positioning.
2. Key details regarding the evaluation testbed are inadequately specified, including sequence lengths, batch sizes, and the distinction between prefill and decode stages. If the current evaluation primarily focuses on single-task autoregressive generation, it remains unclear how the proposed methods would perform in real-world serving scenarios that necessitate dynamic batching of multiple sequences.
3. The configurations and notations presented in Table 1 are somewhat confusing.

---

> ### Author Rebuttal · Authors · 2026-03-30
>
> We thank the reviewer for their supports. We will try to address all the reviewer's questions.
>
> **Q1: How does the proposed method handle concurrent requests in a dynamic batched inference setting?**
> Thank you for the chance to elaborate more. Least-Stale's eviction criterion is batch-size agnostic by design. Staleness is defined at the forward-pass level: an expert is *current* if it was accessed in the current forward pass (or selected by prefetching), and *stale* if it was last accessed in a previous forward pass. This definition does not depend on how many sequences are processed in a pass. In a batched setting, experts activated by any sequence in the batch are naturally marked current and protected from eviction; only experts not accessed by any sequence in the current pass are marked stale. The eviction logic is unchanged regardless of batch size. We will clarify this in the paper.
>
> That said, we intentionally evaluate with batch size 1 to isolate cache policy performance from uncertainty introduced by batching. With larger batches, the union of experts activated across sequences increases cache pressure, and experts needed by one sequence may be evicted due to contention from another. These effects are properties of the workload, not the policy, and would obscure the policy comparison we aim to provide. Batch size 1 allows us to attribute performance differences cleanly to the policy itself.
>
> **Q2: Are there plans to extend SpecMD to support or benchmark against more recent SOTA methods?**
> While we currently do not have plans to extend the framework to support system-level expert offloading, we will add more recent SOTA expert prefetching methods such as a training-free version of Pre-Gated in the camera-ready.
>
> **Q3: Could you clarify the specific experimental configurations (sequence length, batch size, prefill vs. decoding)?**
> We will add a complete evaluation details table to Section 5.1:
> - Input sequence length: up to 512-1024 tokens (task-dependent prompt length)
> - Generation length: up to 1024 tokens
> - Batch size: 1 (single-sequence autoregressive)
> - TTFT measures end-to-end latency including prefill; Token/s measures decoding throughput
> - Decoding uses greedy sampling
>
> We agree these details belong in the main paper and will add them.

---

> > ### Author Rebuttal · Reviewer_YGdn · 2026-04-03
> >
> > Thank you for the response. It addresses part of my concerns. However, I still find several issues insufficiently addressed.
> >
> > 1. Regarding the batch-size-1 evaluation, I understand the intent to isolate policy behavior. Still, I do not think this is sufficient to justify excluding more realistic serving conditions. Dynamic batching and cross-request contention are central to real-world inference systems, and their interaction with cache/offloading policies is precisely what determines practical usefulness. If the method is only validated in single-sequence autoregressive generation, then the scope of the paper should be framed much more narrowly.
> >
> > 2. On Q2, the response remains somewhat vague. It is unclear whether “adding more recent SOTA expert prefetching methods” means adding actual implemented baselines and quantitative comparisons, or expanding the discussion. If the intention is only to expand the discussion, it would be helpful to clarify what specific aspects will be discussed and how that discussion will meaningfully position the paper relative to recent work.

---

> > > ### Author Response · Authors · 2026-04-03
> > >
> > > **Follow-up to Point 1**
> > >
> > > Thank you for the follow-up. We have since run experiments at batch sizes 4, 8 on OLMoE-1B-7B (GSM8K, full dataset, on 5\% cache capacity). The **policy rankings are preserved** across all batch sizes:
> > >
> > > | Config | Batch=1 | Batch=4 | Batch=8 |
> > > |--------|---------|---------|---------|
> > > | Config#1 | 0.677 | 0.660 | 0.655 |
> > > | Config#2 | 0.080 | 0.048 | 0.024 |
> > > | Config#3 | 0.592 | 0.390 | 0.379 |
> > > | Config#4 | 0.664 | 0.663 | 0.657 |
> > > | Config#5 | 0.675 | 0.657 | 0.637 |
> > >
> > > **Follow up on Point 2**
> > >
> > > To clarify: the four papers the reviewer cited (MoE-Lightning, Klotski, MoE-Gen, Pre-Gated Moe) are all **system-level throughput works** targeting multi-batch high-throughput serving. None of them define a training-free, drop-in eviction or prefetch policy that could be instantiated within SpecMD's framework for quantitative comparison. Therefore, "adding more recent SOTA methods" in our camera-ready means **expanded related work discussion**, and not additional baseline.
> > >
> > >  Specifically, we will add a paragraph in Section 2 that explicitly positions each of these four works relative to SpecMD — clarifying that they operate at the scheduling/pipeline layer rather than the per-device cache policy layer that SpecMD studies. We believe this positioning will more clearly delineate SpecMD's contribution and its relationship to the broader literature.

---

### Official Review · Reviewer_gGcS · 2026-03-10

**Soundness:** 3
**Presentation:** 3
**Significance:** 2
**Originality:** 1
**Overall Recommendation:** 2
**Confidence:** 4

**Summary:**

This paper proposes SpecMD, a framework for benchmarking diverse expert caching policies in MoE inference. Along with the framework, the authors introduce Least-Stale, an eviction policy designed to exploit the deterministic expert access patterns inherent in MoE architectures.

**Compliance With Llm Reviewing Policy:**

Affirmed.

**Key Questions For Authors:**

I appreciate the authors' effort in providing a high volume of experimental results. While these data points may be useful for some practitioners, I have concerns regarding the paper's novelty and the selection of baselines.

Expert caching is a well-studied topic, and several existing works have already introduced sophisticated prefetching and caching mechanisms [1][2]. However, this paper seems to omit a deeper discussion or direct comparison with some of these established approaches. Consequently, the evaluation relies heavily on weak baselines like LRU. In practice, I believe MoE inference systems rarely use simple LRU because the deterministic, layer-by-layer access pattern is a fundamental characteristic of the architecture. The paper presents the exploitation of this pattern as a novel contribution, but I find this observation rather trivial, as it is already widely utilized in existing MoE optimization systems.

Lastly, while the details are not included, it seems like only single-batch scenario is considered in the paper. Including more practical scenarios may improve its relevance.

[1] He et al., "ExpertFlow: Optimized Expert Activation and Token Allocation for Efficient Mixture-of-Experts Inference", DAC'25
[2] Hwang et al., "Pre-gated MoE: An Algorithm-System Co-Design for Fast and Scalable Mixture-of-Expert Inference", ISCA'24

**Strengths And Weaknesses:**

Strengths

- The paper includes a wide variety of experimental studies.

Weaknesses

- There is insufficient coverage of closely related prior works that have already addressed expert prefetching and caching.

- The performance comparison relies on too weak baselines, such as LRU.

- The core observations regarding MoE access patterns are not new.

- Experimental studies seem to be assuming only single-batch scenario.

---

> ### Author Rebuttal · Authors · 2026-03-30
>
> We thank the reviewer for their review.
>
> **W1: On missing related works**
>
> We thank the reviewer for pointing out ExpertFlow (He et al., DAC'25) and Pre-gated MoE (Hwang et al., ISCA'24).
>
> - **Pre-gated MoE (ISCA'24)**: Modifies an existing MoE model to activate experts of layer n from layer n-1, with the exception of layer 0. This has the benefit of allowing expert fetches for layer n to overlap with GPU compute of layer n-1, achieving a 100% hit rate. However, the downside is that the MoE must be fully fine-tuned with the modification, thereby creating a new model. This is difficult when the training data for target MoE models is unknown, making it impossible to replicate baseline performance.
>
> - **ExpertFlow (DAC'25)**: Trains a small predictor model that predicts all experts needed across all layers for a given input, and further accelerates throughput with token re-batching for expert efficiency.
>
> While these two works are impressive, we do not consider them to be within our scope. SpecMD specifically scopes to training-free, drop-in **policies** that require no model fine-tuning, no custom hardware, and no retraining, which could be highly useful for end-users without access to expensive training infrastructures.  We will make note of them in our related work, and perhaps add a training-free version of Pre-gated MoE for comparison.
>
> **W2: On weak baselines and trivially-known layer-wise patterns**
>
>
> We appreciate the reviewer's input.
> The main contribution of this paper as we mentioned in the joint response, is not Least Stale, rather it is the unified framework for study MoE caching policies. Least Stale is merely a product of our analysis using this tool. In addition, we believe Least Stale deserve merit on its own.
> While the problem may seem simple, we disagree that Least Stale is trivial, and from the best of our knowledge **no published eviction policy exploits it**:
> - Eliseev uses LRU
> - HOBBIT uses LHU
> - Zhu et al. use score-based eviction
>
> From the two works reviewer cited:
> - Pre-Gated MoE uses LRU/LIFO/LFU
> - ExpertFlow does not utilize an eviction policy, but rather loads all predicted-active experts into cache in a pre-pass and corrects them later if wrong.
>
> To our knowledge, the most popular serving engines for LLMs (vLLM, SGLang) do not currently support MoE expert caching, and therefore do not define an eviction policy.
>
> In any case, Least-Stale provides a straightforward rule-based approach that solves an apparent but overlooked problem. We also note that Reviewer 3zk8 explicitly acknowledges the novelty of Least-Stale in Strength S3: *"The design of the Least-Stale policy is well-motivated... the proposed heuristic effectively resolves the fundamental flaws of prior eviction policies."*
>
> **W3: On single-batch setting**
>
> Our target deployment is single-device inference on memory-constrained hardware (consumer GPUs, edge devices with limited VRAM). We follow standard evaluation setup in most policy papers in the space and only evaluate with a batch size of 1 to isolate cache policy performance free from potential randomness that is inherent when batch size is more than one.
>
>
> **Q1: Expert caching is well-studied — why are stronger baselines not included?**
> Our baseline comparison covers all published training-free, drop-in expert caching systems we are aware of. The two works cited by the reviewer (ExpertFlow, Pre-gated MoE) both require training and are thus outside our scope. We will add discussion of both in the related work section. If the reviewer can identify a specific training-free eviction policy, we welcome the citation and would gladly add it to our benchmarks.

---

> > ### Author Rebuttal · Reviewer_gGcS · 2026-04-04
> >
> > I thank the authors for their detailed responses. However, I have decided to maintain my negative rating based on the following two primary concerns.
> >
> > First, I remain skeptical about the significance of the single-batch setting in real-world systems. To my knowledge, MoE models are rarely deployed in edge settings. Furthermore, the use of an A100 with memory limitations does not accurately reflect the constraints of real mobile devices or consumer-grade GPUs, which would be the real target hardware for such a scenario.
> >
> > Second, I think being 'training-free' does not sufficiently differentiate this work from existing literature. Given that the existing works require only a simple, one-time training phase, which is not a significant burden for most practitioners, the comparative advantage of the proposed method seems to be difficult to demonstrate.

---

### Official Review · Reviewer_3zk8 · 2026-03-11

**Soundness:** 2
**Presentation:** 3
**Significance:** 2
**Originality:** 3
**Overall Recommendation:** 4
**Confidence:** 4

**Summary:**

SpecMD presents a standardized benchmarking framework for evaluating expert caching policies in Mixture-of-Experts (MoE) inference systems. The authors identify a key mismatch between conventional temporal-locality caching assumptions (LRU, LFU) and the deterministic, sequential layer-wise expert access patterns of MoE models. Using the SpecMD framework, they conduct systematic experiments across four MoE architectures (OLMoE-1B-7B, Mixtral-8x7B, Phi-3.5-MoE, Qwen1.5-MoE) under three hardware constraint regimes (1%, 5%, 25% cache capacity at 5 GB/s bandwidth). Based on their empirical findings, the authors propose Least-Stale, an eviction policy that partitions cached experts into 'stale' (from previous forward passes) and 'current' queues, prioritizing eviction of stale experts to prevent collision misses. Experiments show Least-Stale reduces collision misses by up to 85x over LRU and achieves 10.7-44.1% TTFT reductions as a drop-in replacement in existing systems. Additionally, the authors find that score-based prefetching outperforms top-k approaches in most settings despite lower prediction accuracy.

**Compliance With Llm Reviewing Policy:**

Affirmed.

**Final Justification:**

The paper addresses an important and timely systems problem: how to benchmark and improve expert caching policies for MoE inference under hardware constraints. I find the overall research question meaningful and practically relevant, and I view the paper as having solid originality in how it structures the policy space. In particular, SpecMD provides a clean and systematic decomposition of routing, prefetching, eviction, and miss handling, which makes the design space easier to study in a principled way. I also find Least-Stale to be a well-motivated outcome of this framework, and the reported drop-in improvements suggest that the framework can indeed generate practically useful insights. Overall, I view the paper as having good originality and reasonable significance.

My original concerns were mainly about presentation and empirical soundness: the paper’s framing made it somewhat unclear whether the primary contribution was the benchmarking framework or the derived Least-Stale policy; I was also concerned about the statistical support of Table 1, the choice of Config#5 as the representative configuration, and the absence of validation beyond the software-limited A100 setup.

The rebuttal addressed a meaningful portion of these concerns and moved my evaluation in a positive direction. In particular, the clarification that the 100-sample budget applied to the policy sweeps rather than Table 1, together with the additional multi-run statistics, substantially resolved my concern about the statistical stability of the quality results. I also appreciate the authors’ clarification that the intended contribution hierarchy is “Framework → Findings → Policy,” which helps make the paper’s narrative more coherent. In addition, I appreciate the authors’ willingness to revise the presentation around Config#5 and to acknowledge the limitation of the current hardware emulation setup more explicitly.

Some concerns do remain. In particular, I still think the lack of real-device validation limits how broadly the results should be interpreted, and I would have preferred a more principled quantitative justification for highlighting Config#5 as representative. However, after reading the rebuttal, I no longer view these issues as undermining the core value of the paper. Rather, they mainly limit the generality and polish of the current version. I do not view the framework-level conclusions as invalid; instead, I view them as useful first-order insights obtained in a controlled setup, with some caution needed when extrapolating to broader deployment claims.

Taking both the paper and the rebuttal together, my final position is borderline but overall on the positive side. The paper has a clear research motivation, a reasonably original and well-structured framework contribution, and a practically meaningful derived policy. The rebuttal resolved one of my major soundness concerns and improved my confidence in the empirical results. While I still see some limitations in external validity and presentation, I believe these are better characterized as revision issues than as fatal flaws.

**Key Questions For Authors:**

1. What is the paper’s primary contribution: SpecMD as a benchmarking framework or Least-Stale as a new eviction policy?

2. Can you report confidence intervals, repeated-run statistics, or larger-sample results for Table 1 to show that the configuration rankings are stable?

3. Can you validate key claims on at least one real constrained deployment target (not only software emulation on A100)?

4. Why is Config#5 the representative mixed-policy setting?

**Limitations:**

No. Limitations are not adequately discussed. Please add a dedicated section covering: (i) emulation-vs-real-hardware gap; (ii) limited statistical confidence due to small evaluation samples.

**Strengths And Weaknesses:**

Strengths:

S1. [Significance] The proposal of the research question is fully reasonable. MoE expert caching is an increasingly important practical challenge as MoE models proliferate. The observation that temporal locality is fundamentally violated in MoE inference is insightful and strongly supported empirically.

S2. [Originality] The framework design is standardized and logically rigorous. SpecMD cleanly decomposes the caching policy space into four orthogonal dimensions (routing, prefetching, eviction, miss handling) and evaluates them systematically.

S3. [Originality&Soundness] The design of the Least-Stale policy is well-motivated. By integrating staleness with spatial awareness, the proposed heuristic effectively resolves the fundamental flaws of prior eviction policies. The drop-in improvement results presented in Table 2 convincingly demonstrate generalizability the generalization ability of this strategy across diverse baseline systems.


Weaknesses:

W1. [Presentation&Significance] There is a noticeable mismatch between the paper’s claimed contribution and its empirical emphasis. While the paper presents SpecMD as a standardized benchmarking framework for studying MoE caching policies and explicitly lists framework-level contributions, the technical development and experimental narrative are much more centered on Least-Stale. In particular, the paper devotes a dedicated section to Least-Stale and prioritizes validating it in the main evaluation. By contrast, the standalone value of SpecMD as a benchmarking framework—e.g., why it is necessary, what unique capabilities it provides beyond prior evaluation setups, and how it enables fair, comprehensive, and reproducible comparisons—is not demonstrated with the same level of depth. As a result, the paper currently reads more like a Least-Stale paper enabled by SpecMD, rather than a framework/benchmark paper whose primary contribution is SpecMD itself.

W2. [Soundness] The experimental breadth is stronger than the experimental depth. Only 100 examples per task are used for quality evaluation. For tasks like TruthfulQA and NaturalQuestions, accuracy variance over 100 samples can be substantial. No confidence intervals or standard deviations are reported, making it difficult to assess statistical significance of quality differences observed in Table 1.

W3. [Soundness] The choice of Config#5 as the paper’s representative configuration is not fully convincing. As shown in Table 1, Config#5 does not consistently achieve the best TTFT or Token/s across models, yet it is highlighted as the chosen configuration on the basis of “balancing performance and overall speed.” The paper does not provide a sufficiently rigorous justification for this choice. As a result, it is unclear whether Config#5 is truly a principled representative choice or simply a subjective compromise point.

W4. [Soundness] Hardware emulation validity. All experiments use a software-limited A100 GPU with simulated bandwidth (5 GB/s). Real memory-constrained deployments (consumer GPUs, mobile/edge devices) have different memory hierarchy characteristics, thermal throttling, and CPU-GPU interconnect behavior. The degree to which software emulation faithfully represents actual target hardware is unvalidated.

---

> ### Author Rebuttal · Authors · 2026-03-30
>
> We thank the reviewer for their feedback.
> **Q1: What is the paper's primary contribution — SpecMD or Least-Stale?**
> First, we would like to reiterate that our intent for SpecMD is to be a framework that enables easy comparison between MoE expert cache policies. To focus our efforts, we have chosen four broad policy categories to support, i.e., routing, prefetching, eviction, and miss handling. We want to show that by having a common playground to test these policies, engineers and practitioners can draw new conclusions and ideas.
> Therefore, SpecMD has always been our primary contribution; Least-Stale is its most direct output—a policy *derived from* SpecMD's systematic study. We will revise the introduction and abstract to make this hierarchy explicit ("Framework → Findings → Policy")
>
>
> **Q2: Can you report confidence intervals or repeated-run statistics for Table 1?**
>
> We want to clarify a possible confusion. The 100-sample budget remark applies only to the **policy sweep** figures (Sections 5.1–5.3), where we explore a large combinatorial space (~hundreds of policy configurations). Running full-dataset evaluation for each configuration would be computationally prohibitive. **Table 1 (our main results) uses the full dataset minus the 100 samples used for sweep**—GSM8K, TruthfulQA, and NaturalQuestions. We will make this explicit in Section 5.1 and the table caption to avoid ambiguity.
>
> Furthermore, to address your concerns fully, we collected multi-run statistics for the rebuttal on all five configurations in Table 1 for OLMoE-1B-7B and Qwen1.5-MoE-A2.7B. The standard deviations are uniformly small (0.000–0.013), confirming that the rankings are stable:
>
> **OLMoE-1B-7B**
>
> | Config | GSM8K | TruthfulQA | NaturalQA |
> |--------|-------|------------|-----------|
> | Config#1 | 0.677 ± 0.000 | 0.556 ± 0.003 | 0.346 ± 0.004 |
> | Config#2 | 0.080 ± 0.004 | 0.486 ± 0.013 | 0.160 ± 0.003 |
> | Config#3 | 0.592 ± 0.005 | 0.538 ± 0.004 | 0.294 ± 0.008 |
> | Config#4 | 0.664 ± 0.000 | 0.552 ± 0.002 | 0.334 ± 0.003 |
> | Config#5 | 0.675 ± 0.006 | 0.559 ± 0.008 | 0.311 ± 0.004 |
>
> **Qwen1.5-MoE-A2.7B**
>
> | Config | GSM8K | TruthfulQA | NaturalQA |
> |--------|-------|------------|-----------|
> | Config#1 | 0.506 ± 0.000 | 0.617 ± 0.004 | 0.473 ± 0.003 |
> | Config#2 | 0.009 ± 0.003 | 0.506 ± 0.009 | 0.408 ± 0.002 |
> | Config#3 | 0.210 ± 0.003 | 0.606 ± 0.003 | 0.465 ± 0.005 |
> | Config#4 | 0.296 ± 0.000 | 0.617 ± 0.005 | 0.464 ± 0.002 |
> | Config#5 | 0.480 ± 0.003 | 0.609 ± 0.005 | 0.473 ± 0.004 |
>
> We will add these standard deviations to Table 1 in the revision.
>
> **Q3: Can you validate on at least one real constrained deployment target?**
> Thank you for the question. In the paper. we make no claim of hardware parity. Rather, SpecMD controls capacity and bandwidth to emulate two of the biggest constraints on device. We acknowledge that on real devices there are other constraints such as memory hierarchy, thermal throttling, etc., but SpecMD is to faciliate high-level exploration of various MoE execution policies fast with the first order system performance factors. Hence, TTFT and throughput numbers obtained in this setup may not be reflective of real-device behavior (for example in a mobile SoC), but strong early indicator of the expect system performace. Nevertheless, this is a valid limitation we will explicitly acknowledge in the paper.
>
> **Q4: Why is Config#5 the representative configuration?**
> We thank the reviewer for this feedback. We agree that in the current presentation, Config#5 is similar to the other four configs and experiences a degree of quality/speed tradeoff depending on the model being evaluated. We will modify how Config#5 is positioned in our revision to reflect this.

---

> > ### Author Rebuttal · Reviewer_3zk8 · 2026-04-02
> >
> > Thank you for the detailed rebuttal. My concern about the statistical support of Table 1 is largely resolved: I appreciate the clarification that the 100-sample budget applies to the policy sweeps rather than Table 1, and the additional multi-run statistics help strengthen confidence in the stability of the reported quality results. My concern about contribution framing is also partially addressed by clarifying the intended hierarchy of “Framework → Findings → Policy.”
> >
> > However, two core concerns remain. First, the hardware-emulation issue is still largely unresolved. I appreciate the authors’ clarification that they do not claim hardware parity, but the main conclusions are still derived from a software-limited A100 setup without validation on a real constrained deployment target. As a result, the external validity of the reported policy rankings and system trade-offs remains uncertain. Second, the concern about Config#5 is only partially addressed. Repositioning it in the revision helps, but the rebuttal does not yet provide a principled quantitative criterion for why this configuration should be treated as representative.

---

### Decision · Program_Chairs · 2026-04-30

**Decision:**

Accept (regular)

**Comment:**

This paper presents SpecMD, a standardized framework for benchmarking MoE expert caching policies, and uses this framework to derive Least-Stale, a simple eviction policy that yields strong empirical gains under constrained cache settings. Overall, the reviewer discussion confirms that the problem is timely and relevant, and two reviewers viewed the paper positively throughout, highlighting both the breadth of the empirical study and the practical effectiveness of Least-Stale.

During the discussion, the authors addressed a substantial part of the soundness concerns. In particular, they clarified that Table 1 uses the full evaluation sets, not the 100-sample sweep budget, and they provided additional multi-run statistics with very small standard deviations, which largely resolves the concern that the reported quality results might be unstable. They further added follow-up experiments at batch sizes 4 and 8 showing that the policy rankings are preserved, which weakens the objection that the conclusions are purely an artifact of the batch-size-1 setting.

Reviewer 3zk8 acknowledged that the statistical concern was largely resolved and that the framing issue was at least partially addressed, while still maintaining reservations about the lack of real-device validation and the representativeness of Config #5. I consider these remaining concerns legitimate, but they mainly limit the breadth of the claims rather than invalidate the core framework-level conclusions. Reviewer gGcS raised stronger objections about novelty, baseline strength, and the value of the training-free setting. However, these criticisms remained only weakly substantiated in the discussion. In particular, the claim that the core observations are already widely used in practice was not supported with concrete evidence, and the rebuttal explained why the reviewer’s cited systems fall outside the paper’s stated scope of training-free, drop-in policies. This point is also relevant because reviewer 3zk8 explicitly described Least-Stale as well motivated and effective at addressing shortcomings of prior eviction policies.

I also note that the two most negative reviewers did not engage with my follow-up questions asking them to clarify or substantiate their remaining concerns. In particular, they did not respond to requests to explain whether the lack of real-device validation alone should drive rejection, whether the current setup invalidates the core conclusions, whether batch-size-1 evaluation is in fact non-standard in this literature, or to provide quantitative support for the claim that a one-time training phase is not a meaningful burden in practice. Because of that lack of engagement, I assign limited additional weight to those unresolved assertions.

Overall, I find that the paper makes a useful and sufficiently novel contribution. It provides a systematic benchmarking framework for an important systems problem, produces actionable insights about MoE cache-policy design, and introduces a simple policy that appears effective across several models and settings. The remaining weaknesses---especially the absence of real-device validation and the need to sharpen the framing and scope---are real, but they seem more appropriate for revision than for rejection. I therefore recommend acceptance.